# Development and Clinical Validation of a Skin Test for In Vivo Assessment of SARS-CoV-2 Specific T-Cell Immunity

**DOI:** 10.3390/v17091186

**Published:** 2025-08-29

**Authors:** Tikhon V. Savin, Vladimir V. Kopat, Elena D. Danilenko, Alexey A. Churin, Anzhelika M. Milichkina, Edward S. Ramsay, Ilya V. Dukhovlinov, Andrey S. Simbirtsev, Areg A. Totolian

**Affiliations:** 1Saint Petersburg Pasteur Institute, Mira St. 14, 197101 St. Petersburg, Russia; amilichkina@pasteurorg.ru (A.M.M.); simbas@mail.ru (A.S.S.); totolian@pasteurorg.ru (A.A.T.); 2Department of Immunology, Pavlov First State Medical University of St. Petersburg, L’va Tolstogo St. 6-8, 197022 St. Petersburg, Russia; 3LLC “ATG Service Gene”, Maly Prospekt of Vasilievsky Island 57, Building 4, 199178 St. Petersburg, Russia; kopat@service-gene.ru (V.V.K.); dukhovlinov@gmail.com (I.V.D.); 4Institute of Medical Biotechnology, State Research Centre of Virology and Biotechnology “Vector”, Khimzavodskaya St. 9, 633009 Berdsk, Novosibirsk Region, Russia; danilenko_ed@vector.nsc.ru; 5Goldberg Research Institute of Pharmacology and Regenerative Medicine, Tomsk National Research Medical Center, Lenin Avenue 3, 634028 Tomsk, Russia; churin_aa@pharmso.ru

**Keywords:** SARS-CoV-2, cell-mediated immunity, skin test, DTH, CorD_PS, CoronaDerm-PS

## Abstract

A novel skin test for an in vivo assessment of SARS-CoV-2-specific T-cell immunity was developed using CoronaDermPS, a multiepitope recombinant polypeptide encompassing MHC II–binding CD4+ T-cell epitopes of the SARS-CoV-2 structural proteins (S, E, M) and full length nucleocapsid (N). In silico epitope prediction and modeling guided antigen design, which was expressed in *Escherichia coli*, was purified (>95% purity) and formulated for intradermal administration. Preclinical evaluation in guinea pigs, mice, and rhesus macaques demonstrated a robust delayed type hypersensitivity (DTH) response at optimal doses (10–75 µg), with no acute or chronic toxicity, mutagenicity, or adverse effects on reproductive organs. An integrated clinical analysis included 374 volunteers stratified by vaccination status (EpiVacCorona, Gam-COVID-Vac, CoviVac) prior to COVID-19 infection (Wuhan/Alpha, Delta, Omicron variants), and SARS-CoV-2–naïve controls. Safety assessments across phase I–II trials recorded 477 adverse events, of which >88% were mild and self-limiting; no severe or anaphylactic reactions occurred. DTH responses were measured at 24 h, 72 h, and 144 h post-injection by papule and hyperemia measurements. Overall, 282/374 participants (75.4%) exhibited a positive skin test. Receiver operating characteristic analysis yielded an overall AUC of 0.825 (95% CI: 0.726–0.924), sensitivity 79.5% (95% CI: 75.1–83.3%), and specificity 85.5% (95% CI: 81.8–88.7%), with comparable diagnostic accuracy across vaccine, and variant subgroups (AUC range 0.782–0.870). CoronaDerm-PS–based skin testing offers a simple, reproducible, and low-cost method for qualitative evaluation of T-cell–mediated immunity to SARS-CoV-2, independent of specialized laboratory equipment (Eurasian Patent No. 047119). Its high safety profile and consistent performance across diverse cohorts support its utility for mass screening and monitoring of cellular immune protection following infection or vaccination.

## 1. Introduction

COVID-19 infection caused by *Betacoronavirus pandemicum* (SARS-CoV-2) rapidly developed into a global pandemic, seriously affecting public health and socioeconomic systems worldwide [1]. Studies conducted during the pandemic have shown that the virus disrupts the normal functioning of immune cells. This can lead to an imbalance in the immune response and uncontrolled inflammatory processes, especially in patients with severe disease [2]. During primary infection, SARS-CoV-2 binds to cellular receptors, primarily ACE2 [3,4], which initiates a complex chain of events from viral entry into cells to the activation of both innate and adaptive immunity [5,6,7]. Such pathogenetic features complicate timely recognition of the virus by the immune system and contribute to the development of severe clinical conditions [8,9].

The basis of the cellular immune response to viral infections is the interaction of T-cell receptors (TCRs) with peptides presented by molecules of the major histocompatibility complex (MHC) [10]. In COVID-19, this process becomes especially important, since specific CD4+ and CD8+ T cells play a decisive role in controlling the viral infection, while simultaneously forming long-term immunological memory [11,12,13]. Studies have shown that early activation of the virus-specific T-cell response correlates with a favorable clinical prognosis, while its delay can lead to the progression of severe forms of illness [14,15]. In the latter, various T-lymphocyte phenotype patterns are observed, as well as a significant decrease in their overall number (lymphopenia). This reflects the multifactorial impact of the virus on the immune system [16,17,18,19].

One of the most important components of adaptive immunity are virus-specific memory T cells, which remain in the body after infection or vaccination, providing long-term protection [20,21,22,23,24,25]. With COVID-19, there is a rapid induction of such cells. Their specificity is determined by the expression level of viral proteins, primarily structural antigens (S, M, N, E), which are the dominant targets for cytotoxic and helper T lymphocytes [11,12,13,26]. Along with the cellular response, humoral immunity is formed, reflected in the synthesis of specific antibodies (mainly IgG, IgM, IgA) [27,28,29,30]. However, the correlation between antibody concentrations and specific T-cell levels is weak. This highlights the need to develop additional methods for assessing cellular immunity [27].

Current methods for assessing the T-cell immune response in COVID-19 include the use of ELISPOT, intracellular cytokine staining (ICS), and interferon-γ release assays (IGRA) [31,32,33,34]. Despite the high accuracy of these methods, their widespread implementation is limited by high cost, long analysis times, and availability issues (specialized equipment, expensive reagents). An alternative method for assessing specific cellular immunity to various infectious pathogens are skin tests, such as the tuberculin test [35]. Existing skin tests based on assessing the delayed-type hypersensitivity reaction are a promising direction for rapid diagnostics, but they also have limitations associated with the spectrum of antigens used [36,37,38]. This approach has also been used for COVID-19.

In December 2020, the CoviDCELL^®^ skin test containing 25 µL (0.1 mg/mL) of reconstituted lyophilized SARS-CoV-2 RBD was tested for the first time in Spain to diagnose the T-cell response after COVID-19 vaccination. Limited studies have shown nearly 100% positive results in vaccinated individuals, including patients with congenital immune deficiencies and immunosuppression after kidney transplantation [39,40,41,42]. On 14 December 2021, it was announced that the TNX-2100 skin test was approved (US FDA) for a phase I clinical trial, but study results have not yet been published. The test is comprised of three different mixtures of synthetic peptides (TNX-2110, -2120, -2130) representing different SARS-CoV-2 protein components. The aforementioned initiatives highlight the relevance of developing alternative methods for determining specific T-cell immunity, preferably enabling routine screening at minimal cost.

Based on published data on COVID-19 immunopathogenesis and the role of virus-specific memory T cells in protecting the human body from infection, we developed the CoronaDerm-PS preparation. It is a recombinant polypeptide containing SARS-CoV-2 structural protein antigens (S, M, N, E) to assess specific T-cell immunity to SARS-CoV-2 [43]. In order to confirm the safety and specific activity of the developed method, several studies were performed: preclinical study; clinical trials (phases I-II); and two cohort studies.

## 2. Materials and Methods

### 2.1. Development of the Recombinant Antigen Preparation CoronaDerm-PS

To construct the recombinant antigen, immunogenic T-cell CD4+ epitopes were determined for SARS-CoV-2 structural protein (S, N, M, E) sequences in silico [44]. To predict/identify high-affinity viral epitopes and MHC-II binding, the TepiTool [45] and NetMHCIIpan [46] algorithms were used. These were based on the “panel of 27 most frequent A and B alleles” approach, with the inclusion of specific alleles (HLA-DR, HLA-DP, HLA-DQ) [47,48]. In the “prediction method” (IEDB), a “moderate number of peptides” was selected, and the default epitope length was 15 amino acids (aa). Peptides were predicted based on an IC50 cutoff value of ≤1000 nM. In NetMHCIIpan, binding of 15 aa peptides to 27 MHC-II alleles was also predicted. The 0.5% frequency epitopes were selected based on strongest binding prediction. All 15 aa peptides were predicted to bind to 27 MHC class II molecules, which account for 97% of the HLA-A and HLA-B allelic variants in most ethnic groups [47]. In the next step of antigen development, selected CD4+ T-cell epitopes were searched in the ViPR database (https://www.viprbrc.org, accessed on 31 July 2025). Refinement parameters included ‘family Coronaviridae’, ‘human host’, and ‘experimentally determined T-cell epitopes’. Modeling of T-cell receptor binding to the selected epitopes and formation of the epitope/MHC-II complex was performed using ERGO algorithms, which are applicable to both CD4+ and CD8+ T-cell epitopes [49].

In developing our antigen design, we constructed several virtual variants of recombinant coronavirus antigens. Due to its conservation and immunogenicity, full-length SARS-CoV-2 N protein was considered first. Next, selected CD4+ T-cell antigenic determinants of structural proteins (S, E, M) were added. The structure of the constructed recombinant proteins was modeled using I-TASSER. Stereochemical parameters of the model antigens were analyzed using Ramachandran plots via the RAMPAGE server [50]. Physicochemical parameters of the proteins were analyzed using the ProtParam server [51]. Isoelectric points were calculated using the ‘Protein isoelectric point calculator’ (http://isoelectric.org/, accessed on 15 April 2021) [52]. After combining the aforementioned methods, and assessing the immunogenicity of the antigens in silico, a hybrid sequence of the recombinant coronavirus antigen was selected. Next, the obtained aa sequence (containing the most immunogenic regions of the SARS-CoV-2 structural proteins) was aligned using BLAST [53] against human proteins. This was to ensure that there was no similarity with human protein sequences and to avoid the occurrence of autoimmune reactions upon antigen introduction.

The antigen sequence was synthesized and cloned into the pET24a(+) vector. The resulting pCorD_PS plasmid was sequentially transformed into *Escherichia coli* DH5α, then into Rosetta (DE3) strains. The producer strain for recombinant *Escherichia coli* CorD_PS antigen was used for further antigen production by biotechnological methods. Subsequent steps included chromatographic purification in three stages (ion exchange, hydrophobic interaction, size exclusion), quality control, and formulation of the CoronaDerm-PS preparation, as described previously [43].

### 2.2. Preclinical Study Phase in Animals

Preclinical studies of CoronaDerm-PS were carried out to assess the safety and specific activity of the preparation. These were conducted following approvals from the relevant authorities: Institutional Ethics Committee (protocol PI-01/2021), State Research Center of Virology and Biotechnology VECTOR, Rospotrebnadzor (No. 2, 11 May 2021); and the Institutional Ethics Committee, Russian Institute of Medical Primatology (No. 84, 24 March 2022). These were conducted using: 393 guinea pigs (*Cavia porcellus*); 485 laboratory mice (Table 1) of various lines (outbred CD-1, BALB/c, C57Bl/6); and seven clinically healthy, sexually mature rhesus macaque (*Macaca mulatta*) males. The primates (weighing from 8800 to 11,750 g) were grouped as follows: six males previously vaccinated with Gam-COVID-Vac in one group; and one male as control.

Preclinical experiments were conducted in animals. Such animal studies provide the most complete information on the toxicity and specific activity of a substance that is intended to be used by humans. Animals were kept in separate rooms and under conditions complying with: rules adopted by the European Convention for the Protection of Vertebrate Animals (Strasbourg, 1986); Principles on Good Laboratory Practice (OECD, ENV/MC/CUEM (98)17, 1997); and federal standards (GOST 33215-2014). Control of environmental parameters included maintaining temperature (20–24 °C), humidity (45–65%), and natural light, with daily monitoring of condition. Standard granulated feed for rodents (GOST R 51849-2001) was given ad libitum in the feeding cavity of the steel lattice lid of the cage. Laboratory-quality water was given ad libitum in standard autoclaved bottles with steel spout lids. Animals were adapted in the vivarium for 7 days before the start of administration. During this period, animals were examined daily for external condition and clinical signs before randomization. Animals were randomly distributed into groups using body weight as a criterion such that individual weights did not differ by more than 10% from the average weight of animals of the same sex. Each animal was assigned an individual number (marked with a color code on the body). The number of animals used in the study was sufficient for complete registration of the effects studied. No animal was excluded after randomization. There was no blinding during the experimental or measurement phases. The procedure order and measurement times were standardized to reduce bias, and a matched control group was used. The preparation was administered in the morning from 10:00 to 11:00, and the order of animals in each session was random. Humane termination criteria were not established insofar as the intervention was short and minimally invasive. In the study, the CoronaDerm-PS preparation was administered subcutaneously (Table 2). Animals in the control group received a saline solution in an equivalent volume.

Mortality, serious conditions, general health, and signs of toxicity were assessed once a day or twice a day (morning, afternoon) in all groups throughout the observation period. Examination was performed in a cage or on an open surface. During examination, the following features and/or organs were assessed: behavioral features (increased, decreased activity); gait (muscle tone, tremor, balancing); temperament (lethargy, excitability, aggressiveness); appearance (emaciation, obesity); coat condition (hair loss, protruding hair, spotted, dull, smooth, shiny); eyes (lacrimation, inflammation, corneal opacity); ears (inflammation, color—pale or red, secretion, sensitivity, twitching); limbs (color, swelling); and teeth (color, loss). Physiological functions assessed included breathing (rate, character, features), salivation (consistency, amount), urination (color, amount), and excrement (color, consistency). Body weight was recorded in all groups before administration and weekly during the experiment (8, 15 days after administration).

To assess the immune response, a delayed-type hypersensitivity test was used. The preparation was administered intradermally to guinea pigs and rhesus macaques. After 72 h, the local reaction (erythema, infiltration) was assessed on a 5-point scale: (0) no visible reaction; (1) pale pink erythema over the entire area or its periphery; (2) bright pink erythema over the entire area or its periphery; (3) red erythema over the entire area; (4) infiltration and edema of the skin (thickening of the skin fold) with or without erythema; or (5) erythema, severe infiltration, focal ulcerations (necrosis), with possible hemorrhages and crusting.

To assess acute toxicity, the preparation was administered subcutaneously once (25 μg protein/animal) at a dose of 0.5 mL to mice and 1 mL to guinea pigs. After 1 and 7 days, data were obtained by the following: hematological analysis; blood biochemical analysis; and clinical observations. Animals were euthanized by CO_2_ inhalation followed by exsanguination. Pathomorphological examination of internal organs was performed after euthanasia. During necropsy, the following were examined: external condition of the body; internal surfaces and passages; cranial cavity; thoracic cavity; abdominal and pelvic cavities and their organs/tissues; neck and its organs/tissues; and musculoskeletal system. The following internal organs were weighed: liver, heart, lung, kidneys, adrenal glands, spleen, and thymus. Paired organs were weighed together. Samples of the following organs were taken for histological examination: brain, bone marrow, lungs, liver, heart, kidneys, adrenal glands, stomach, small intestine, large intestine, reproductive organs, thymus, spleen, injection site, and regional lymph nodes. Sections were examined using light microscopy with an assessment and description of tissue condition.

To assess chronic toxicity, the preparation was administered subcutaneously daily for 10 days (10 μg/animal). Observation was carried out for 17 days, taking into account body weight, temperature, and clinical signs of toxicity. Clinical blood analysis and biochemical parameters were assessed on the 1st and 7th days after completion of administration. After euthanasia, a histological examination of the organs was carried out. Local irritant action was assessed with a single administration by studying skin reactions in mice and guinea pigs. For the repeated administration regime, daily subcutaneous injection was carried out for 10 days. Skin samples and regional lymph nodes were studied histologically after 1 and 7 days after completion of preparation administration.

To determine the immunological safety of the preparation, various parameters were studied: effects on the thymus or spleen weight and cellularity; peritoneal macrophage phagocytic activity; the number of antibody-forming cells; and antibody titers. To assess allergenic properties, sensitization of guinea pigs was carried with repeated administrations (0.8 μg/kg dose). Anaphylactic properties were determined on the 14th day by intracardiac administration of the preparation; the degree of allergic reaction was assessed according to the Weigle scale. Evaluation of mutagenicity was carried out using the Ames test and chromosomal aberration analysis. The Ames test used indicator strains of *Salmonella typhimurium* (TA98, TA100, TA1535, TA1537), and the preparation was tested at five concentrations (maximum 51 μg/mL). The number of revertant colonies was estimated 72 h after start of incubation.

The genotoxic effect of CoronaDerm-PS on somatic cells was determined by the method of chromosomal aberration count in murine bone marrow cells. The study was conducted in two stages. In the first stage, 15 male mice were used (5 per group). Preparations were administered once subcutaneously at a dose recommended for use in humans, equal to 10 μg/mouse, in a volume of 0.2 mL. Saline solution was administered in a similar manner in the same volume. Cyclophosphamide solution was administered intraperitoneally (20 mg/kg) in a volume of 0.2 mL/mouse.

In the second stage, CoronaDerm-PS preparation or saline solution were administered four times via the following regime: once per day subcutaneously at a dose of 10 μg/mouse (male, female) in a volume of 0.2 mL. The exposure time was 24 h. Approximately 1.5–2 h before the end of exposure, colchicine solution (0.04%) was administered intraperitoneally to animals (0.2 mL/20 g animal body weight) to accumulate metaphase chromosomes. With animals euthanized by instantaneous dislocation of the cervical vertebrae, bone marrow was washed out of the femur, and preparations of metaphase plates (chromosomes) were prepared using the standard method. Potential mutagenic effect was assessed by the presence of chromosomal abnormalities in metaphases, taking into account structural aberrations. For analysis, 100 cells were selected from each animal sample at the metaphase stage. Chromosomal aberrations (fragments, exchanges) were analyzed in each metaphase cell.

Analysis of any effects of the preparation on the reproductive system was carried out with guinea pigs as part of experiments on acute and chronic toxicity. The weight and morphology of the testes and ovaries were estimated.

For hematological analysis, blood was collected from the tail vein of mice, or from the marginal ear vein of guinea pigs. The Mythic18 (vet) automatic hematological analyzer (ORPHEE SA, Plan-les-Ouates, Geneva, Switzerland) was used. The number of erythrocytes, hemoglobin levels, leukocytes, platelets, and white blood cell count were assessed. For biochemical analysis, blood was taken from the heart of mice, or from the cardiac vein of guinea pigs. The concentrations of ALT, AST, ALP, glucose, total protein, albumin, cholesterol, triglycerides, creatinine, and urea were assessed using an XL-100 automatic analyzer (Erba Mannheim, Brno, Czech Republic). For histological analysis, organs were fixed, dehydrated, and embedded in paraffin. Staining was performed with hematoxylin and eosin. Studies were performed using a Mikmed-5 light microscope (LOMO, Saint Petersburg, Russia) with a description of tissue changes.

### 2.3. Evaluation of Safety and Specific Activity of CoronaDerm-PS in Volunteer Groups

#### 2.3.1. Demographic and Clinical Characteristics of Volunteer Groups

As part of an integrated analysis of the safety and specific activity of CoronaDerm-PS, 374 volunteers were included. They were divided into four main groups. Group 1 was comprised of conditionally healthy individuals, without a history of COVID-19 or SARS-CoV-2 vaccination, identified in phase I of the clinical trial (protocol No. CD-PS-01/21). The female/male ratio (f/m) was 11/9 (*n* = 20). The age range was 19 to 76 years $\bar{x}$ = 36.86 ± 14.84).

Group 2 was comprised of those previously vaccinated against SARS-CoV-2 (14–180 days before inclusion in the study). Its subgroups were as follows. Group 2a featured those vaccinated with EpiVacCorona (peptide-based vaccine): the f/m ratio was 45/34 (*n* = 79); the age range was 18 to 79 years ($\bar{x}$ = 44.27 ± 16.73). Group 2b featured those vaccinated with Gam-COVID-Vac (vector-based vaccine): the f/m ratio was 43/39 (*n* = 82); the age range was 18 to 81 years ($\bar{x}$ = 43.79 ± 16.12). Group 2c featured those vaccinated with CoviVac (inactivated vaccine): the f/m ratio was 18/7 (*n* = 25); the age range was 22 to 60 years ($\bar{x}$ = 36.80 ± 11.05).

Group 3 was comprised of those who had recently experienced COVID-19 (14–180 days before inclusion in the study). Its subgroups were as follows. Group 3a featured those who had COVID-19 caused by the Wuhan variant or the Alpha variant: the f/m ratio was 19/14 (*n* = 33); the age range was 24 to 79 years ($\bar{x}$ = 46.91 ± 14.37). Group 3b featured those who had COVID-19 caused by the Delta variant: the f/m ratio was 50/30 (*n* = 80); the age range was 20 to 92 years ($\bar{x}$ = 44.99 ± 15.88). Group 3c featured those who had COVID-19 caused by Omicron subvariants: the f/m ratio was 36/16 (*n* = 52); the age range was 21 to 71 years ($\bar{x}$ = 41.85 ± 13.28).

Group 4 was comprised of individuals without a history of COVID-19 or SARS-CoV-2 vaccination identified in clinical trial phases I-II (protocol No. CD-PS-01/21). The f/m ratio was 11/12 (*n* = 23). The age range was 19 to 76 years ($\bar{x}$ = 37.87 ± 15.18).

Data for the integrated analysis were obtained from phases I-II of the clinical trial (Groups 1, 2, 3b, 4) conducted according to protocol No. CD-PS-01/21: single-blind, placebo-controlled trial approved by the Russian Ministry of Health Ethics Council (protocol No. 298, dated 18 January 2022) with permission of the Russian Ministry of Health (No. 90, dated 10 February 2022). Data were also obtained via a cohort study (Group 3a,c) approved by the local ethics committee of the Saint Petersburg Pasteur Institute (extract from protocol No. 99, dated 24 September 2024). The size of an adequate sample was calculated mathematically by formula (Appendix B). The inclusion and exclusion criteria are provided in the Appendix A.

#### 2.3.2. Skin Test Assessment and Interpretation Criteria

To assess specific activity, a skin test was performed with the CoronaDerm-PS preparation, which permits the detection of formed cellular immunity to SARS-CoV-2. The preparation was administered intradermally in the forearm area. Results were assessed dynamically 24 h (±2 h), 72 h (±3 h), and 144 h (±4 h) after administration.

Assessment was carried out by measuring the transverse size of the infiltrate (papule) and the hyperemia zone using a transparent ruler. The following classifications were used to interpret skin reactions. ‘Negative reaction’ was defined as follows: complete absence of infiltrate or hyperemia; or the presence of only a “prick reaction” (slight redness at the needle puncture site). ‘Inconclusive reaction’ was defined as follows: the presence of hyperemia without infiltrate. ‘Positive reaction’ was defined as follows: presence of an infiltrate (papule) of any size, taking into account hyperemia. ‘Positive reactions’ were further classified by severity: ‘mild’ (infiltrate < 5 mm); ‘moderate’ (infiltrate ≥ 5 mm, but <10 mm); ‘severe’ (infiltrate ≥ 10 mm, but <15 mm); or ‘hyperergic reaction’ with infiltrate (≥15 mm) and/or the presence of vesicular-necrotic changes, lymphangitis, or lymphadenitis (regardless of infiltrate size). A feature of CoronaDerm-PS is the possibility of a false-positive reaction, expressed as short-term hyperemia, which occurs immediately after administration and disappears after 48–72 h.

#### 2.3.3. Clinical and Laboratory Investigations

Clinical blood analysis included counts (RBC, WBC, platelet), hemoglobin level, and leukocyte formula. Analysis was performed on an ABX Pentra 60 automatic analyzer (Horiba Ltd., Kyoto, Japan). The erythrocyte sedimentation rate (ESR) was determined using the Panchenkov method. Biochemical blood analysis was performed to assess organ function (liver, kidneys) and carbohydrate metabolism. Activities (ALT, AST, LDH, alkaline phosphatase) and levels (total protein, bilirubin, cholesterol, glucose, creatinine, urea, C-reactive protein) were determined. Immunoglobulin E levels were assessed to identify possible allergic reactions. Analyses were performed on the CA-270 Clinical Chemistry Analyzer (Furuno Electric Co., Ltd., Nishinomiya, Hyogo, Japan) using the turbidimetry method with DiaS reagents (DIAKON-DS, JSC, Pushchino, Moscow Region, Russia).

Evaluation of the blood coagulation system included determination of the international normalized ratio (INR), activated partial thromboplastin time (aPTT), and fibrinogen level using corresponding reagents (RPA RENAM, Moscow, Russia). Serological studies were performed to detect antibodies to the HIV and hepatitis B/C viruses by ELISA using the CombiBest anti-HIV-1,2 kits (AO Vector-Best, Novosibirsk, Russia) and using ARCHITECT hepatitis determination reagents (Abbott Laboratories, Abbott Park, IL, USA). To determine specific antibodies to SARS-CoV-2, the N-CoV-2-IgG PS kit (Saint Petersburg Pasteur Institute, Saint Petersburg, Russia) was used. General urine analysis included various parameters (color, specific gravity, pH, protein, glucose, leukocytes, erythrocytes, bilirubin, ketones, nitrites, urobilinogen) using the “dry chemistry” method on the Clinitek Status Plus analyzer (Siemens Healthcare Diagnostics GmbH, Forchheim, Bavaria, Germany). Determination of viral RNA in nasopharyngeal swabs was carried out by the RT-PCR method using the RIBO-prep reagent kit (Central Research Institute of Epidemiology, Moscow, Russia) and the “COVID-2019 Amp” kit for SARS-CoV-2 RNA detection (Saint Petersburg Pasteur Institute, Saint Petersburg, Russia).

Determination of T-lymphocyte subpopulations was carried out using CD45-FITC (Becton Dickinson, Franklin Lakes, NJ, USA), CD3-APC (Beckman Coulter, Inc., Brea, CA, USA), CD4-PC7 (Beckman Coulter, Inc., Brea, CA, USA), and CD8-PerCP (Becton Dickinson, Franklin Lakes, NJ, USA) labeling reagents. The FACSCanto II flow cytometer (Becton Dickinson, Franklin Lakes, NJ, USA) and reagents were used following generally accepted gating tactics and manufacturer instructions.

Assessment of IFN-γ production by CD4+ T lymphocytes was carried out according to the following study protocol. Venous blood from donors was collected in test tubes with heparin (10 U/mL). Blood was mixed with sterile phosphate-buffered saline (PBS) in a 1:2 ratio and layered on a density gradient of 1.077 g/mL Histopaque-1077 (Sigma-Aldrich (Merck), St. Louis, MO, USA), followed by centrifugation for 30 min at 400 g (18-22 °C). Following centrifugation, the mononuclear cell layer formed at the phase boundary was collected. The resulting cell suspension was washed twice with complete culture medium (CCM) for 7 min at 300g. The composition of CCM was RPMI-1640 (Biolot, Saint-Petersburg, Russia) supplemented with 10% inactivated fetal bovine serum (FBS, Biolot, Saint Petersburg, Russia), 50 μg/mL gentamicin (Biolot, Saint Petersburg, Russia), and 2 mM L-glutamine (Biolot, Saint Petersburg, Russia). The number of obtained cells was then determined using a hemocytometer.

To set up the experiments, 200 μL of cell suspension (1–2 × 10^7^ cells/mL) in CCM were added to the wells of 96-well plates (Sarstedt AG & Co. KG, Numbrecht, North Rhine-Westphalia, Germany). Staphylococcus enterotoxin B (SEB) at a final concentration of 1 μg/mL was used as a positive reaction control. An equal volume of the culture medium used to prepare the final SEB solution was used as a negative control for the reaction. CoronaDerm-PS was used to stimulate cells (5 μg/mL final concentration). In order to suppress cytokine secretion by cells, the Golgi apparatus blocker brefeldin-A was added to all samples (10 μg/mL final concentration). Samples were then incubated at 37 °C (5% CO_2_) for 18 h. Following incubation, cells were resuspended in cooled PBS containing 2% FBS, transferred to centrifuge tubes, and washed twice with excess PBS (300 g for 8 min). The resulting cell suspension was used to assess intracellular cytokine production.

Samples were then stained with antibodies to surface antigens (CD45RA-ECD, CD4-PC7, CD3-APC-Cy7). They were also stained with Zombie Aqua dye (BioLegend, San Diego, CA, USA), which enables distinguishing between live and dead cells, according to manufacturer instructions. Further sample preparation was carried out using IntraPrep Permaebilization Reagent (Beckman Coulter, Inc., Brea, CA, USA) for intracellular antigen staining according to manufacturer instructions. To identify cells that have accumulated IFN-γ in the cytoplasmic compartment, antibodies against human IFN-γ conjugated with FITC were used. Upon completion of sample preparation for analysis, cells were resuspended in a 2% neutral formalin solution in PBS.

Samples were analyzed using a NovoCyte Flow Cytometer (Agilent Technologies, Santa Clara, CA, USA) equipped with three diode lasers (405, 488, 638 nm). Each sample was assessed for the relative content of IFNγ+ cells with the CD3+CD4+CD45RA– phenotype. The results were expressed as % IFNγ+ lymphocytes out of the total number of memory T-helper cells (CD3+CD4+CD45RA–).

Instrumental studies included fluorography to exclude tuberculosis, 12-lead ECG, and a pregnancy test (immunochromatographic analysis of urine β-hCG). The analyses conducted in each volunteer group are listed in Table 3.

Adverse events were assessed on a 4-point scale from “none” to “severe” with an analysis of occurrence frequency and an assessment of causal relationship according to World Health Organization (WHO) criteria. Statistical analysis was performed using SPSS Statistics 26 (IBM, Armonk, NY, USA). Methods of descriptive statistics were used to describe demographic and clinical characteristics: quantitative variables with a normal distribution are presented as the mean with standard deviation (M ± SD); categorical variables are presented as absolute values and relative frequencies (n, %). To assess differences between groups by categorical characteristics, the Pearson χ^2^ test was used. The strength of the relationship between categorical variables was additionally assessed using Cramér’s V. Comparison of quantitative variables between two independent groups was performed using the Mann–Whitney test. The prognostic significance of indicators was assessed by ROC analysis. Calculation of confidence intervals for sensitivity/specificity used the Wilson method in the RStudio environment version: 2025.05.1+513 (Posit Software, Boston, MA, USA) using built-in statistical functions. Comparison of area under the curve (AUC) values between groups was performed using the DeLong test. Statistical significance was set at *p* < 0.05.

## 3. Results

### 3.1. Finalized Recombinant Antigen

The recombinant SARS-CoV-2 antigen preparation CoronaDerm-PS was developed to assess specific T-cell immunity in vivo. The structure of the multiepitope construct (recombinant coronavirus antigen) includes the following: MHC-II binding epitopes of structural proteins (S, M, E); and full-length nucleocapsid (N) protein, due to being the most conserved within the Coronaviridae family and one of the most immunogenic (Figure 1) [54].

The developed chimeric protein consists of 486 amino acids. Its other values are the following: molar mass 53,059 daltons; isoelectric point (pI) 9.56; hydrophobicity index (GRAVY) −0.785; aliphatic index 61.91; and extinction coefficient 53,860 (1 mg/mL of protein solution has an optical density of 1.02 at 280 nm). A 3D model of the protein is shown in Figure 2.

The production process used for obtaining the preparation (CoronaDerm-PS) makes it possible to obtain 5–7 g of the coronavirus recombinant antigen (CorD_PS) per one cultivation cycle (30 L fermentation medium) [48]. The process yields recombinant antigen material with favorable qualities: purity > 95%; residual LPS < 800 EU/mg; DNA of the producer strain ≤200.0 ng/mg protein; and host cell proteins (HCP) ≤ 10 μg/mg (≤1% of protein content).

### 3.2. Preclinical Study Results with CoronaDerm-PS in Animals

The specific activity of CoronaDerm-PS was assessed using the delayed-type hypersensitivity test (DTH) in guinea pig (Cavia porcellus) and rhesus macaque (Macaca mulatta) models previously vaccinated against SARS-CoV-2. The preparation was administered intradermally into the forearm area (Figure 3). Development of a local skin reaction was recorded after 24 h (±2 h), 72 h (±3 h), and 144 h (±4 h).

The results showed that in guinea pigs, with subcutaneous sensitization with the preparation in doses from 10 to 75 μg, intradermal administration of the resolving dose caused the development of a skin reaction in 86–100% of animals. The average infiltrate diameters were as follows: 4.44 ± 0.64 mm (M ± SD) at a dose of 10 μg; 6.12 ± 0.53 mm (M ± SD) at a dose of 25 μg; and 8.67 ± 0.46 mm (M ± SD) at a dose of 50 μg.

Increasing the sensitizing dose above 50 μg did not lead to a significant increase in reaction. This indicates that an immune response plateau has been reached. In rhesus macaques, intradermal administration of the preparation at a dose of 0.2 mL (concentration 50 μg/mL) caused the development of an infiltrate with a diameter of 0.8 cm after 72 h, thus indicating the presence of cellular immunity to SARS-CoV-2.

To assess acute toxicity, the preparation was administered subcutaneously once (25 μg/mouse, 50 μg/guinea pig). Animals were observed for 7 days. No signs of toxic effects were detected. Animal behavior, body weight, and appetite did not change relative to the control group. Hematological parameters (leukocyte, erythrocyte, platelet levels) and biochemical parameters (ALT, AST, creatinine, urea etc.) remained within the physiological norm.

Chronic toxicity assessment was performed by daily subcutaneous administration of the preparation for 10 days at doses of 10 μg/mouse and 10 μg/guinea pig. Animals were observed for 17 days with monitoring of clinical condition, body weight, and temperature. No clinical signs of toxicity were detected. Hematological and biochemical parameters remained stable, and histological examination of internal organs did not show signs of inflammation, necrosis, or fibrosis.

To assess the local irritant effect, the preparation was administered once (0.5 mL/mouse, 1.0 mL/guinea pig) or multiple times subcutaneously (0.2 mL/animal). Observation was carried out for 7 days after administration. No clinically significant irritation was recorded with a single administration. Histological examination of the skin revealed weak lympho-macrophage infiltrates that disappeared after 7 days. With repeated administration, no signs of a local inflammatory reaction or necrosis at the injection site were detected.

To assess immunological safety, several parameters were examined: the weights of selected organs (thymus, spleen, lymph nodes); the cellularity of these organs; and phagocyte functional activity. The weight of lymphoid organs did not change compared to the control group of animals. Evaluation of phagocytic activity showed no effect of the preparation on the oxidation–reduction potential of peritoneal macrophages. Number of spleen antibody-form cells and antibody titers to sheep erythrocytes did not change throughout the experiment (Table 4).

Guinea pigs were sensitized with subsequent intramuscular administration of the preparation. Anaphylactic reactions were assessed on the 14th day using the Weigle scale. No anaphylactic reactions were detected. The local skin reaction was mild, which corresponds to accepted standards for immunobiological preparations.

Mutagenicity was assessed using two methods: the Ames test and chromosomal aberration analysis. The Ames test used four *S. typhimurium* strains, and the preparation was tested at five concentrations. After 72 h of incubation, no increase in the number of revertant colonies was detected compared to the control. Chromosomal aberration analysis was performed via administration of the preparation to C57Bl/6 mice by two regimes (single dose, four dose). The number of aberrations in bone marrow did not exceed control values, indicating the absence of a mutagenic effect (Table 5).

Potential effects on the reproductive system were assessed in guinea pigs by acute and chronic toxicity experiments. It was demonstrated that the preparation did not affect the weight of the testes or ovaries. The morphology of the seminiferous tubules, morphology of the spermatogenic epithelium, and the number of follicles at different developmental stages did not differ from control values (Table 6).

Thus, preclinical studies showed that the CoronaDerm-PS preparation: does not cause acute or chronic toxic effects; does not have a mutagenic effect; does not lead to immune system disruptions; and does not have a negative effect on reproductive function. The specific activity of the preparation was confirmed by the development of a skin reaction (DTH) in intradermal animal testing, indicating the formation of cellular immunity. No animal deaths were observed during the experiments. As such, the obtained data made it possible to initiate a study of CoronaDerm-PS safety and specific activity in healthy volunteers.

### 3.3. Integrated Analysis Results of CoronaDerm-PS Safety and Specific Activity in Volunteer Groups

The safety of the preparation was assessed in two clinical trial phases to identify adverse events (AEs) and to assess their relationship with administration of the preparation. In phase I of the clinical trial, 45 AEs were recorded (86.7% mild, 13.3% moderate). All AE resolved on their own or after minimal symptomatic therapy. According to World Health Organization Uppsala Monitoring Centre (WHO-UMC) criteria, nine AEs had a definite relationship with administration of the preparation; one AE was probable; and 24 AEs were assessed as questionable. There were no severe AEs requiring medical intervention or hospitalization (Appendix C).

The second phase involved 269 volunteers divided into five groups based on previous COVID-19 infection and the type of vaccine previously received. During the observation period, 432 AEs were recorded: 383 were mild (88.6%); 41 were moderate (9.5%); and eight were severe (1.9%). Most AEs were recorded within the first 24 h after preparation administration and disappeared on their own within 1–3 days. No cases of anaphylaxis, severe allergic reaction, or other serious adverse events (SAEs) were identified. This confirms the favorable safety profile of the preparation. To control safety, monitoring was conducted: blood analysis (clinical, biochemical); coagulation test; and general urine analysis. Clinical blood analysis showed that the levels of leukocytes, erythrocytes, and platelets remained within reference values. A slight increase in the level of eosinophils and monocytes was observed in 5.2% of volunteers, but these changes did not exceed reference values. The biochemical blood test, coagulation test and general urine test parameters also remained within physiological values, indicating an absence of toxic effects with CoronaDerm-PS.

Specific activity was assessed based on the results of the skin test and laboratory examination. A positive skin test result was recorded in 282 of 374 participants (75.4%) (Figure 4). The distribution of results by group is presented in Table 7.

ROC analysis showed that in the overall sample: AUC = 0.825 (95% CI: 0.726–0.924); SE = 0.0505; *p* < 0.001; sensitivity = 79.5% (95% CI: 75.1–83.3%); and specificity = 85.5% (95% CI: 81.76–88.7%). This indicates a high diagnostic ability of the skin test to detect specific T-cell immunity to SARS-CoV-2. Data were also obtained on the high diagnostic accuracy of the developed skin test in groups vaccinated with various vaccines, as well as in groups that had experienced COVID-19 caused by different genetic variants. The results of statistical processing of skin test results for different groups are presented in Table 8.

In order to compare results in different volunteer groups (assessment of CoronaDerm-PS specific activity), the DeLong test was carried out (Table 8). According to the data, comparable accuracy was confirmed in different subgroups of volunteers. This allows it to be used in mass studies involving individuals with different medical histories.

## 4. Discussion

The CorD_PS recombinant antigen is a chimeric, multiepitope construct including immunogenic regions of all SARS-CoV-2 structural proteins. Its use expands the capabilities of the diagnostic test system. Existing diagnostic reagents often assess antibody (Ab) levels (IgG, IgM) to certain viral elements, such as spike protein or its receptor-binding domain (RBD). However, some vaccines do not contain full-length S protein or the RBD. If a vaccine does not contain (or encode) viral protein fragments to which Abs are formed, existing Ab-based diagnostics will be ineffective. The CorD_PS recombinant antigen enables assessment of the T-cell immune response after vaccination, even after vaccines from various platforms that may not elicit the Abs usually measured in diagnostics. In addition, the antigens previously described in the literature for use in evaluating T-cell immunity to SARS-CoV-2 in vivo contain S protein components. This limits their use in the presence of mutations in the genes encoding this viral protein.

Studies have confirmed the safety of CoronaDerm-PS, both in laboratory animals [55], and in phases I-II of a clinical trial [56]. In a study of the preparation’s safety, the absence of toxic, locally irritating, anaphylactogenic, or mutagenic effects was experimentally demonstrated. A study with volunteers also confirmed the experimental data obtained and showed good tolerability. The adverse events noted in clinical trial phases I-II were mostly mild AEs. These are typical of the method of administration (intradermal) and characteristic of skin tests assessing specific immunity [57]. However, no serious adverse events or fatal outcomes were observed.

The data obtained in the clinical study on the specific activity of CoronaDerm-PS showed high sensitivity and specificity of the skin test in individuals vaccinated against SARS-CoV-2, as follows. Group 2a had 76.60% sensitivity (95% CI: 67.36–83.85%) and 80.00% specificity (95% CI: 71.07–86.69%). Group 2b had 81.70% sensitivity (95% CI: 73.09–88.01%) and 87.00% specificity (95% CI: 79.11–92.20%). Group 2c had 84.00% sensitivity (95% CI: 70.70–91.95%) and 87.00% specificity (95% CI: 74.22–93.96%). High sensitivity and specificity were also seen with individuals who had experienced COVID-19 caused by the Delta genetic variant. Group 3b had 79.70% sensitivity (95% CI: 70.79–86.42%) and 87.50% specificity (95% CI: 79.60–92.62%). A limitation of the clinical study was the small sample in Group 2c. This led to wider confidence intervals for sensitivity and specificity, which are explained by the low availability of the vaccine in civil circulation.

Nevertheless, the data confirm the specific activity of the preparation for assessing immunity after vaccination with different vaccine types (EpiVacCorona, Gam-COVID-Vac, CoviVac). This is associated with the presence of the main SARS-CoV-2 structural antigens in the recombinant chimeric antigen. The structure of CoronaDerm-PS also ensures that key features of the method (sensitivity, specificity) are sustained when assessing post-infectious immunity, regardless of the SARS-CoV-2 genetic variant that caused infection.

Data for Group 3 reflects this, as follows. Group 3a showed 81.80% sensitivity (95% CI: 69.39–89.91%) and 86.95% specificity (95% CI: 75.35–93.56%). Group 3b showed 79.70% sensitivity (95% CI: 70.79–86.42%) and 87.50% specificity (95% CI: 79.60–92.62%). Group 3c showed 76.47% sensitivity (95% CI: 65.49–84.77%) and 86.95% specificity (95% CI: 77.27–92.89%).

A major limitation of our study is relatively small number of volunteers in group 2b. This limitation was caused by limited vaccine availability in the general population, and the fact that the study was conducted in Saint Petersburg, Russia, which restricted the evaluation to a single ethnic group (Eastern/Northern European) and the vaccine types available locally. Future studies should involve larger and more diverse cohorts, including immunocompromised individuals. These studies should also include an assessment of SARS-CoV-2-specific T-cell immunity using complementary laboratory methods, i.e., ELISPOT, IGRA, ICS. This way, data on the diagnostic accuracy of the intradermal test will be more precise. While the test demonstrated high specificity, cross-reactivity with endemic human coronaviruses cannot be completely excluded, which may account for occasional false-positive results. Further investigations are required to evaluate the extent of such cross-reactivity and its impact on test performance. This emphasizes the need for longitudinal studies to assess the durability of IDR test positivity.

Thus, taking into account the safety and specific activity data on the developed method, we can assert the following. The described skin test using the CoronaDerm-PS preparation enables assessment of the presence of both T-cell immunity types (post-vaccination, post-infectious) in individuals, and specialized laboratory equipment is not needed. The method’s simplicity and reproducibility, alongside the clarity of its results, allow this skin test to be used in mass screening studies aiming to personalize preventive vaccination.

## 5. Conclusions

The developed recombinant antigen containing immunogenic CD4+ T-cell epitopes of SARS-CoV-2 structural proteins (S, E, M, N) represents a useful diagnostic preparation. It can be used for qualitative assessment of specific immune protection against coronavirus infection and/or assessment of the immunogenicity of candidate vaccines in clinical trials. Studies have demonstrated that CoronaDerm-PS has a high safety profile. Due to its recombinant chimeric peptide design, it also has high specific activity for determining the presence of cellular immunity to SARS-CoV-2 in relevant groups (previously vaccinated, recovered). Most adverse events were mild, did not require medical intervention, and resolved on their own. The originality and diagnostic relevance of the CoronaDerm-PS skin test are supported by its protection under Eurasian Patent No. 047119, assigned the IPC classifications G01N 33/68 (2006.01), A61B 5/00 (2006.01), C07K 14/165 (2006.01), and C07K 11/00 (2006.01) [58].

The skin test showed high diagnostic accuracy regardless of the vaccine used or the SARS-CoV-2 genetic variant that caused COVID-19. An important advantage of the developed skin test method with the CoronaDerm-PS preparation is its simplicity. An intradermal injection can be performed in a procedure room without the use of specialized equipment or reagents. This makes widespread use in various settings possible. Taken together, the study’s findings allow us to consider the skin test method using the CoronaDerm-PS preparation a promising tool for assessing T-cell immunity to SARS-CoV-2 in clinical practice, applicable for mass screening.

## Figures and Tables

**Figure 1 viruses-17-01186-f001:**
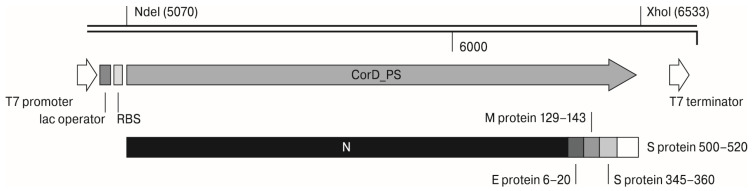
Linear structure of the chimeric protein. The layout is as follows: N protein (full-length) from aa 1–419; E protein epitope from aa 420–434 (aa 6–20); M protein epitope from aa 435–449 (aa 129–143); S protein epitope from aa 450–465 (aa 345–360); S protein epitope from aa 466–486 (aa 500–520). The construct has earlier been described in detail [48]. Numbers in parenthesis indicate protein reference positions.

**Figure 2 viruses-17-01186-f002:**
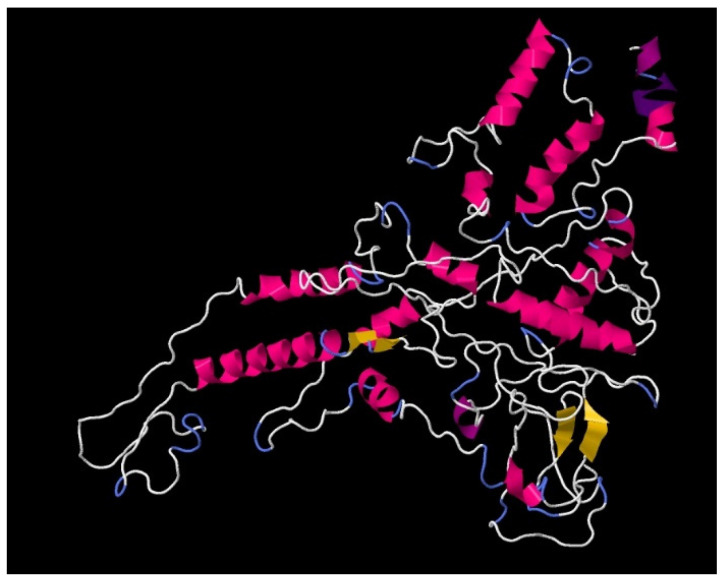
Modeled structure of the recombinant coronavirus antigen. The 3D protein model was rendered with I-TASSER.

**Figure 3 viruses-17-01186-f003:**
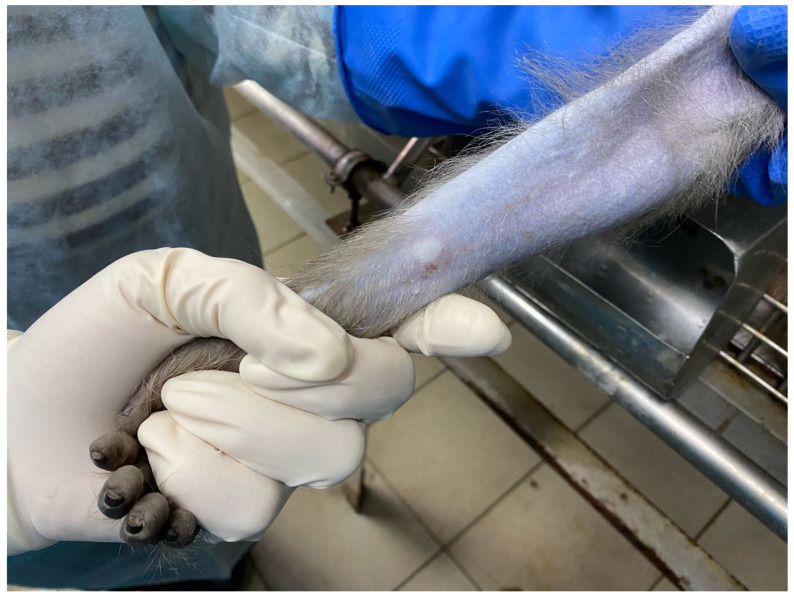
Appearance of the skin test in animal testing (*Macaca mulatta*).

**Figure 4 viruses-17-01186-f004:**
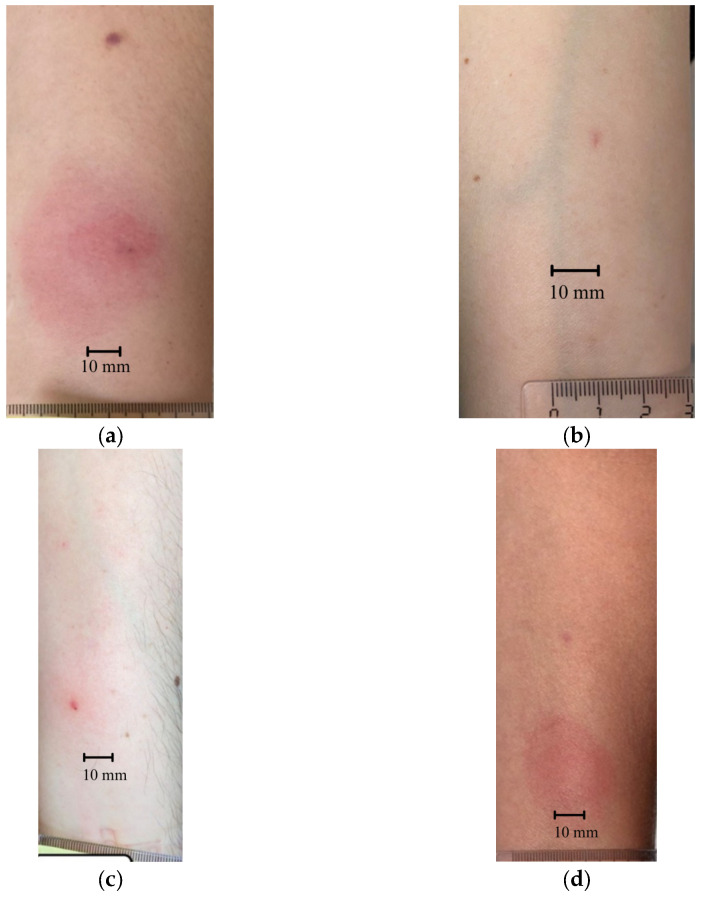
Examples of skin reaction test results: (**a**) ‘positive’; (**b**) ‘negative’; (**c**) ‘false-positive’; (**d**) ’inconclusive’.

**Table 1 viruses-17-01186-t001:** Distribution of laboratory animals involved.

Experiment	Animals
Evaluation of specific activity, CoronaDerm-PS	guinea pigs (*), 46 males and 46 females, weighing 350–400 g, aged 6–7 weeks (*n* = 92) white-sided guinea pigs (**), 14 males and 15 females, weighing 350–400 g, aged 7–8 weeks (*n* = 29)
Assessment of “acute” toxicity, CoronaDerm-PS	outbred CD-1 mice (***), 40 males and 40 females weighing 25–30 g, aged 8 weeks (*n* = 80) shorthaired guinea pigs of motley color (***),40 males and 40 females, weighing 650–750 g, aged 12–14 weeks (*n* = 80)
Assessment of chronic toxicity, CoronaDerm-PS	outbred CD-1 mice (***), 40 males and 40 females, weighing 25–30 g, aged 8 weeks (*n* = 80) shorthaired guinea pigs of motley color (***),20 males and 20 females, weighing 650–750 g, aged 12–14 weeks (*n* = 40)
Assessment of local irritant action, CoronaDerm-PS	outbred CD-1 mice (***), 80 males and 80 females, weighing 25–30 g, aged 8 weeks (*n* = 160) shorthaired guinea pigs of motley color (***),60 males and 60 females, weighing 650–750 g, aged 12–14 weeks (*n* = 120)
Assessment of immunological safety, CoronaDerm-PS	BALB/c mice (****), female, weighing 18–20 g, aged 8 weeks (*n* = 130)
Assessment of allergenic properties, CoronaDerm-PS	guinea pigs (****), 16 males and 16 females, weighing 350–400 g, aged 6-7 weeks (*n* = 32)
Evaluation of mutagenic potential by chromosomal aberration test, CoronaDerm-PS	C57Bl/6 mice (****), males, weighing 22–24 g, aged 9–10 weeks (*n* = 35)
Evaluation of reproductive system effects, CoronaDerm-PS	conducted as part of the study of acute and chronic toxicity in guinea pigs

* Obtained from the Rappolovo Laboratory Animal Nursery, Kurchatov Institute Federal Research Center (Leningrad Region, Rappolovo). ** Obtained from the nursery, Vector Research Center of Virology and Biotechnology, Rospotrebnadzor (Koltsovo, Novosibirsk Region) and Krolinfo, LLC (Novaya Village, Orekhovo-Zuevsky District, Moscow Region). *** Obtained from the Dept. of Experimental Biological Models, E.D. Goldberg Research Institute of Pharmacology and Regenerative Medicine (Tomsk). **** Obtained from the nursery, Vector Research Center of Virology and Biotechnology, Rospotrebnadzor (Koltsovo, Novosibirsk Region).

**Table 2 viruses-17-01186-t002:** Characteristics of the CoronaDerm-PS preparation.

Component	Concentration
Active Ingredient	
Recombinant fusion protein containing SARS-CoV-2 structural proteins (partial)	10.0 ug
Inactive Ingredients	
polysorbate 20	2 uL (1%)
Tris-HCl	0.605 mg (25 mM)
sodium chloride	1.8 mg (0.9%)
phenol	0.3 mg (0.15%)
water for injection	to 0.2 mL
pH	7.4

**Table 3 viruses-17-01186-t003:** Analyses conducted by volunteer group.

Analyses	Group 1	Group 2			Group 3			Group 4
		Group 2a	Group 2b	Group 2c	Group 3a	Group 3b	Group 3c	
receipt of written informed consent	X	X	X	X	X	X	X	X
collection and registration of medical history	X	X	X	X	X	X	X	X
physical examination	X	X	X	X	X	X	X	X
vital sign assessment (BP, HR, RR, temperature)	X	X	X	X	X	X	X	X
blood analysis (biochemical, clinical, coagulogram, total IgE)	X	X	X	X		X		X
serological analysis (HIV, hepatitis B/C, syphilis)	X	X	X	X		X		X
general urine analysis	X	X	X	X		X		X
anti-SARS-CoV-2 IgG determination (ELISA)	X	X	X	X	X	X	X	X
analysis of nasopharyngeal, oropharyngeal swabs for SARS-CoV-2 RNA (PCR)	X	X	X	X		X		X
evaluation of T-cell immunity by flow cytometry (ex vivo)	X	X	X	X		X		X
lung fluorography	X	X	X	X		X		X
pregnancy test	X	X	X	X	X	X	X	X
ECG	X	X	X	X		X		X
CoronaDerm-PS injection	X	X	X	X	X	X	X	X
adverse event assessment	X	X	X	X	X	X	X	X

**Table 4 viruses-17-01186-t004:** Assessment of the immunological safety of CoronaDerm-PS.

Indicator	Experimental Animal Group	
	Control Saline Solution (*n* = 8)	CoronaDerm-PS 10 μg/0.2 mL/Mouse (*n* = 8)
**One day after administration**		
mouse mass, g	19.4 ± 0.3	20.2 ± 0.4
splenic mass index, ×10, mg/g	100.6 ± 3.7	105.1 ± 4.9
bsolute number of nucleated splenocytes, ×10^6^/organ	180.6 ± 18.0	178.8 ± 10.0
thymic weight index, ×10, mg/g	30.3 ± 1.9	31.5 ± 1.5
absolute thymocyte count, ×10^6^/organ	80.7 ± 7.0	75.1 ± 6.8
Ab producing cells/spleen	15,081.6 ± 774.0	14,145.9 ± 1795.2
hemagglutinin titer, log_2_	8.0 ± 0	8.0 ± 0
**Twenty-one days after administration**		
mouse mass, g	22.0 ± 0.4	23.2 ± 0.3 *
splenic mass index, ×10, mg/g	88.3 ± 3.0	98.7 ± 3.7 *
absolute number of nucleated splenocytes, ×10^6^/organ	333.6 ± 19.7	333.0 ± 16.3 *
thymic weight index, ×10, mg/g	27.5 ± 1.9	28.6 ± 2.0
absolute thymocyte count, ×10^6^/organ	85.4 ± 7.6	97.0 ± 3.0 *
Ab producing cells/spleen	22,128.5 ± 1957.8	22,847.3 ± 1688.9
hemagglutinin titer, log_2_	8.0 ± 0	8.0 ± 0

* statistically significant difference compared to the group of animals that were administered saline solution (*p* < 0.05).

**Table 5 viruses-17-01186-t005:** Number of chromosomal aberrations in bone marrow cells of mice after exposure to CoronaDerm-PS.

Experimental Conditions (Frequency of Administration, Preparation, Dose)	Cells Examined	Number of Aberrations				Damaged Cells, %
		Single Fragments, %	Paired Fragments, %	Exchanges, %	Multiple Aberrations, %	
CoronaDerm-PS 10 ug protein/0.2 mL/mouse, once, males	500	0.4	0	0	0	0.4
0.9% NaCl 0.2 mL/mouse, once, males	500	1.0	0	0	0	1.0
CoronaDerm-PS 10 ug protein/0.2 mL/mouse, 4 times, males	500	0.8	0.2	0	0	1.0
0.9% NaCl 0.2 mL/mouse, 4 times, males	500	1.2	0	0	0	1.2
CoronaDerm-PS 10 ug protein/0.2 mL/mouse, 4 times, females	500	0.8	0	0	0	0.8
0.9% NaCl 0.2 mL/mouse, 4 times, females	500	1.4	0	0	0	1.4
Cyclophosphamide 20 mg/kg, 0.2 mL/mouse, once, males	500	13.8	3	0.2	8.0	25.0

**Table 6 viruses-17-01186-t006:** Ovarian mass and mass coefficient after a single subcutaneous administration of CoronaDerm-PS at a dose of 50 ug/1.0 mL/animal (M ± m) in guinea pigs.

Group		Animal Mass (g)	Ovarian Mass (g)	Mass Coefficient (%)	
Day 1	control (*n* = 5)	658.20 ± 53.88	0.11 ±0.01	0.02 ± 0.00	
	1.0 mL/animal (*n* = 5)	621.60 ± 19.76	0.11 ± 0.00	0.02 ± 0.00	
Day 7	control (*n* = 5)	635.00 ± 20.02	0.07 ± 0.01	0.01 ± 0.00	
	1.0 mL/animal (*n* = 5)	652.80 ± 21.70	0.09 ± 0.01	0.01 ± 0.00	

**Table 7 viruses-17-01186-t007:** Skin test results using CoronaDerm-PS in different groups (*n* = 374).

Group	Positive (Volunteers)	Inconclusive (Volunteers)	Negative (Volunteers)
Group 1 (no history of illness or vaccination), phase I	0	0 ^†^	20
Group 2a (EpiVacCorona)	61	0 ^†^	18
Group 2b (Gam-COVID-Vac)	67	0 ^†^	15
Group 2c (CoviVac)	21	0 ^†^	4
Group 3a (Wuhan strain and Alpha variant)	27	5	1
Group 3b (Delta variant)	63	0 ^†^	17
Group 3c (Omicron subvariants)	40	7	5
Group 4 (no history of illness or vaccination), phases I, II	3	0 ^†^	20

^†^ In clinical trial phases I-II, inconclusive results were taken to be ‘negative’.

**Table 8 viruses-17-01186-t008:** ROC analysis by group.

Group	AUC (95% CI)	SE	*p*	Sensitivity (95% CI)	Specificity (95% CI)
Group 2a (EpiVacCorona)	0.782 (0.678–0.887)	0.053	<0.001	76.60% (95% CI: 67.36–83.85%)	80.00% (95% CI: 71.07–86.69%)
Group 2b (Gam-COVID-Vac)	0.843 (0.751–0.935)	0.047	<0.001	81.70% (95%CI: 73.09–88.01%)	87.00% (95% CI: 79.11–92.20%)
Group 2c (CoviVac)	0.87 (0.764–0.975)	0.054	<0.001	84.00% (95% CI: 70.70–91.95%)	87.00% (95% CI: 74.22–93.96%)
Group 3a (Wuhan strain and Alpha genetic variant)	0.844 (0.733–0.955)	0.057	<0.001	81.80% (95% CI: 69.39–89.91%)	86.95% (95% CI: 75.35–93.56%)
Group 3b (Delta variant)	0.844 (0.760–0.928)	0.043	<0.001	79.70% (95% CI: 70.79–86.42%)	87.50% (95% CI: 79.60–92.62%)
Group 3c (Omicron subvariants)	0.799 (0.689–0.909)	0.056	<0.001	76.47% (95% CI: 65.49–84.77%)	86.95% (95% CI: 77.27–92.89%)
DeLong test, Group 2	2a vs 2b	Z = 0.861	*p* = 0.39		
	2a vs 2c	Z = 1.162	*p* = 0.246		
	2b vs 2c	Z = 0.377	*p* = 0.706		
DeLong test, Group 3	3a vs 3b	Z = 0	*p* = 1		
	3a vs 3c	Z = 0.563	*p* = 0.574		
	3b vs 3c	Z = 0.637	*p* = 0.524		

## Data Availability

Not applicable.

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
