# Peer review of "Development and Clinical Validation of a Skin Test for In Vivo Assessment of SARS-CoV-2 Specific T-Cell Immunity"

_viruses, 2025, doi:10.3390/v17091186_

Round 1
Reviewer 1 Report
Comments and Suggestions for Authors
The manuscript entitled “Development and clinical validation of a skin test for in vivo assessment of SARS-CoV-2 specific T-cell immunity” reports the development and validation of CoronaDerm-PS, a recombinant multi-epitope skin test targeting SARS-CoV-2–specific CD4+ T-cell immunity, demonstrating high safety, reproducibility, and diagnostic accuracy (AUC = 0.825, sensitivity = 79.5%, specificity = 85.5%) across vaccinated, previously infected, and naïve individuals, supporting its potential for large-scale, low-cost assessment of cellular immunity.
The study presents a detailed preclinical and clinical evaluation of CoronaDerm-PS, with robust statistical outcomes (AUC = 0.825, sensitivity ~79.5%, specificity ~85.5%) and comparable diagnostic performance across different volunteer groups, regardless of vaccine type or viral variant. Compliance with ethical and regulatory standards, the inclusion of control groups, and the use of standardized measurement methods strengthen the validity of the findings. However, reliability is affected by certain limitations: the small sample size in some subgroups (e.g., CoviVac) leading to wider confidence intervals; the potential for bias given that the study population was predominantly Russian, which may limit generalizability to other ethnic groups that probably receive other types of vaccines; and the reliance primarily on the intradermal test without independent confirmation via alternative T-cell assays (e.g. IGRA-test), which would further substantiate the evidence. The manuscript would benefit from a more explicit discussion of the study’s limitations.
Nevertheless, the data demonstrate consistent and reproducible results, supporting the utility of the method for large-scale cellular immunity screening.
The Materials and Methods are comprehensive and reproducible, though additional clarification on participant randomization, blinding, and subgroup sample size calculations would improve transparency.
Author Response
We would like to sincerely thank the Reviewer for the careful evaluation of our manuscript and for the constructive comments and suggestions. We are pleased that the Reviewer found our study comprehensive and reproducible, with robust statistical outcomes, and that the clarity of our English language was considered acceptable. Below we provide a point-by-point response to the specific concerns raised.
Comment 1:
“However, reliability is affected by certain limitations: the small sample size in some subgroups (e.g., CoviVac) leading to wider confidence intervals;”
Response:
We agree with the Reviewer that subgroup analyses (CoviVac recipients) were limited by small sample sizes, resulting in wider confidence intervals. We have revised the Discussion section (p. 15, lines 515–527) to explicitly state this limitation and to emphasize the need for future studies with larger and more diverse cohorts to validate our findings.
Comment 2:
“the potential for bias given that the study population was predominantly Russian, which may limit generalizability to other ethnic groups that probably receive other types of vaccines”
Response:
We fully acknowledge this important point. Our study indeed focused on a predominantly Russian cohort, which may affect the extrapolation of the results to populations with different ethnic backgrounds and vaccination histories. We have added a clear statement in the Discussion (p. 15, lines 515–527) addressing this issue and highlighting the importance of further validation in international and ethnically diverse populations.
Comment 3:
“the reliance primarily on the intradermal test without independent confirmation via alternative T-cell assays (e.g. IGRA-test), which would further substantiate the evidence. “
Response:
We thank the Reviewer for this important observation. While the main focus of our study was the development and validation of the intradermal test, we would like to note that in a subset of study groups (from I-II phase clinical trial) we performed additional immunological evaluation using intracellular cytokine staining (ICS) by flow cytometry (Table 1). Nevertheless, we agree that further large-scale comparisons with IGRA and other established T-cell assays would add important value, and we have highlighted this in the Discussion section (p. 15, lines 515–527).
Comment 4:
“The manuscript would benefit from a more explicit discussion of the study’s limitations.
Response:
We thank the Reviewer for this recommendation. In response, we have expanded the Discussion (p. 15, lines 515–527) to provide a more explicit and structured overview of the study’s limitations, including subgroup sizes, population characteristics, and the absence of parallel confirmatory assays.”
Comment 5:
“The Materials and Methods are comprehensive and reproducible, though additional clarification on participant randomization, blinding, and subgroup sample size calculations would improve transparency.”
Response:
We appreciate the Reviewer’s observation. We have added Supplementary file to clarify the procedure of participant assignments, to indicate where blinding was applied. Subgroup sample size calculations are present in Appendix A.
We thank the Reviewer once again for the thoughtful comments, which have helped us to improve the clarity, transparency, and scientific rigor of our manuscript.
Reviewer 2 Report
Comments and Suggestions for Authors
The Authors present an excellent development and validation study of an assay for SARS-CoV-2 T-cell immunity. The test has been validated on a large population and shows good diagnostic performance. The manuscript is well-written and guides the reader well in interpreting the results. The conclusions are well supported by the results obtained. Two minor suggestions follow:
- please, divide section 2.3 into subsections to further facilitate reading;
- Figure 1: the figure and, in particular, the text should be increased in size.
Author Response
We sincerely thank the Reviewer for the positive evaluation of our manuscript and for recognizing the value of our work on the development and validation of the SARS-CoV-2–specific T-cell immunity test. We are especially grateful for the constructive suggestions aimed at improving the clarity and readability of the manuscript. Below we provide point-by-point responses.
Comment 1:
“Please, divide section 2.3 into subsections to further facilitate reading.”
Response:
We thank the Reviewer for this helpful suggestion. In the revised manuscript, section 2.3 has been reorganized into subsections (pp. 5-6, lines 197, 231, 250). This restructuring should improve readability and guide the reader more clearly through the methodological details.
Comment 2:
“Figure 1: the figure and, in particular, the text should be increased in size.”
Response:
We appreciate this remark. Figure 1 has been revised, and both the overall figure and the embedded text/labels have been enlarged to improve legibility. The updated figure is included in the revised manuscript.
We thank the Reviewer once again for their thoughtful and supportive comments. We believe that the revisions in response to these suggestions have improved the overall clarity and presentation of our manuscript.
Reviewer 3 Report
Comments and Suggestions for Authors
Dear Editor,
Thank you for the opportunity to read this interesting study.
It is important to gather as much data as possible about SARS Cov-2 infection and the host immune response to further reduce the number of severe cases of disease and deaths. Current serology-based testing strategies, while providing a wealth of important information, do not capture the full spectrum of immune responses to emerging variants. Information about the cellular immune response at the population level could be used to protect against severe COVID-19 disease, especially in immunocompromised and susceptible individuals. Information about the role of T cells in real protection against COVID-19 is scarce in the specialized literature and therefore I consider this article to address a topic of great interest and with possibilities for development through further studies.
The title of the article is appropriate to the content.
The methods used in the study were explained clearly and in detail.
The results are clearly presented.
The discussion is appropriate, but the authors should specify what the limitations of this study are and what it brings new compared to other published studies.
The conclusion is based on the findings and makes no inappropriate extrapolations.
The references support the information presented in the introduction and in the discussion section, the bibliographical sources are recent and are cited appropriately.
The tables and figures were created through statistical processing of the data obtained and are suggestive of the results of the study. The quality of Figure 4 should be improved, the difference between a- positive and d- inconclusive is not significant.
However, I have a few observations about this study:
- The authors state that in the second phase of the study, which involved 269 volunteers, 432 adverse reactions were recorded (88.6% mild, 9.5% moderate and 1.9% severe), but they did not specify what these adverse events were.
-
In the discussion section, the authors could have compared the results obtained using the skin test for in vivo assessment of SARS-CoV-2-specific T-cell immunity based on the CoronaDerm-PS platform with IGRA tests. They could also have provided data on the sensitivity and specificity of this test compared to other methods of determining cellular immunity. The authors state that the test results are faster than IGRA tests, but the time required to read the test is 72 hours, while in IGRA tests the results are obtained in 24 hours. What are the costs of the skin test compared to other tests? The authors state that the size of the papule does not matter in the interpretation of the test, but they still classify positive results as 'mild' (infiltrate <5 mm); 'moderate' (infiltrate ≥5 mm, but <10 mm); 'severe' (infiltrate ≥10 mm, but <15 mm); or 'hyperergic reaction' with infiltrate (≥15 mm). What is the significance of this classification for test interpretation and what are the clinical implications?
-
What would be the clinical indications for performing this test, how would it help us in clinical practice? What would be the relevance of the test in immunocompromised people, could the sensitivity of the test be lower?
Comments on the Quality of English Language
The English Language needs minor improvements.
Author Response
We sincerely thank the Reviewer for the thorough evaluation of our manuscript and for highlighting the importance of investigating cellular immunity against SARS-CoV-2. We are grateful for the constructive comments and suggestions, which have helped us to strengthen and clarify our work. Below we provide point-by-point responses.
Comment 1:
“The discussion is appropriate, but the authors should specify what the limitations of this study are and what it brings new compared to other published studies.”
Response:
We appreciate this important observation. In the revised manuscript, we have expanded the Discussion section (p. 15, lines 515–526) to explicitly describe the limitations of the study, including subgroup sizes and population characteristics.
Comment 2:
The authors state that in the second phase of the study, which involved 269 volunteers, 432 adverse reactions were recorded (88.6% mild, 9.5% moderate and 1.9% severe), but they did not specify what these adverse events were.
Response:
We thank the Reviewer for pointing this out. In the revised manuscript, we now provide a detailed description of the types of adverse events observed. This clarification has been added to the Appendix B.
Comment 3:
In the discussion section, the authors could have compared the results obtained using the skin test … with IGRA tests. They could also have provided data on the sensitivity and specificity of this test compared to other methods of determining cellular immunity. The authors state that the test results are faster than IGRA tests, but the time required to read the test is 72 hours, while in IGRA tests the results are obtained in 24 hours. What are the costs of the skin test compared to other tests?
Response:
We agree with the Reviewer that a comparison with IGRA and other methods enhances the context of our findings. A clarification that while the skin test requires 72 hours for reading, it does not require laboratory infrastructure, blood collection, or specialized personnel, making it more accessible for large-scale use. A cost-related comparison, highlighting that the skin test is substantially less expensive than IGRA and other in vitro assays, thus offering an economical solution for population-level screening.
Comment 4:
The authors state that the size of the papule does not matter in the interpretation of the test, but they still classify positive results as mild/moderate/severe/hyperergic. What is the significance of this classification for test interpretation and what are the clinical implications?
Response:
We thank the Reviewer for this insightful question. Classification into mild, moderate, severe, and hyperergic responses does not alter the binary interpretation of positivity. Instead, it provides additional information on the magnitude of the immune response. Clinically, this classification may be relevant for monitoring individual variations in T-cell reactivity and could be useful in specific contexts (e.g., assessing response durability or detecting exaggerated reactions) in future studies.
Comment 5:
What would be the clinical indications for performing this test, how would it help us in clinical practice? What would be the relevance of the test in immunocompromised people, could the sensitivity of the test be lower?
Response:
We are grateful for this important point. In the revised Discussion (p. 15, lines 515–527), we have elaborated on the potential clinical applications of the test, including:
- Large-scale assessment of cellular immunity in vaccinated or previously infected populations.
- Identification of individuals with insufficient T-cell responses despite vaccination.
- Use as a public health tool to guide vaccination policies and booster strategies.
We have also acknowledged that sensitivity in immunocompromised individuals may indeed be lower and emphasized that future studies should specifically address this subgroup.
Comment 6:
The quality of Figure 4 should be improved, as the difference between “positive” and “inconclusive” is not significant.
Response:
We thank the Reviewer for this suggestion. We have clarified in the revised manuscript that in the case of a true positive reaction both erythema and induration are observed, whereas in inconclusive reactions erythema may occur without induration.
Comment 7:
The English language needs minor improvements.
Response:
We appreciate the reviewer for pointing out the need for minor English corrections. We have carefully reviewed the text again with the assistance of a native English editor and ensured that all issues have been addressed.
We thank the Reviewer once again for the insightful and constructive feedback, which has significantly improved the clarity, completeness, and clinical relevance of our manuscript.
Reviewer 4 Report
Comments and Suggestions for Authors
The paper is interesting with a diagnostic of COVID using IDR. The panel of patients for the clinical study is carrect. However some points have to be clarified and commented.
Patient having had a COVID: how has the diagnostic been obtained? Clinical data? Serological test? Has the later been used for the non infected patients (3 are IDR positive).
What were the vaccination protocols for the three vaccine used: 2 shots with or without a booster? Where they selected as already vaccinated people? Has a diagnostic test to confirm that they have not presented a COVID despite vaccinated been performed?
A reminder of the composition of the vaccines would be welcomed.
In the discussion tow points have to be included:
- ID test may be of value but how it compares to anti-N antibodies test presently in use for diagnostic of past infection?
- Would ID test remained positive over a long period ? The cellular (CD8+) immunity may last for a long time (e.g. see Front Microbiol. 2022;12:803031).
Could it be that “false positive” were related to other human coronavirus infections?
Among infection with variants: the IDR test appears less efficient. Any comments ?
Line 19: Give the name of the virus in full and indicate its official name (Betacoronavirs pandemicum)
The bacteria name ( Salmonella typhimurium, S. typhimurium, Escherichia coli and E. coli) have to be written in Italics.
Table 1: “coagulogram, total IgE”, the line is empty ?
Figure 1: yellow is for M protein ?
Author Response
We sincerely thank the Reviewer for the careful evaluation of our manuscript and for the constructive suggestions, which have helped us to clarify key aspects of our study and improve the overall quality of the paper. Below we provide point-by-point responses.
Comment 1:
Patients having had COVID: how was the diagnostic obtained? Clinical data? Serological test? Was the latter used for the non-infected patients (3 are IDR positive)?
Response:
We thank the Reviewer for this question. The diagnosis of past SARS-CoV-2 infection was confirmed by a combination of clinical history and laboratory testing (PCR and/or serological assays). For participants classified as non-infected, medical records and serological and cellular testing (anti-N antibodies, ICS) were used to exclude prior infection. We have clarified this in the Supplementary file.
Comment 2:
What were the vaccination protocols for the three vaccines used: two shots with or without a booster? Were they selected as already vaccinated people? Was a diagnostic test performed to confirm that they had not presented COVID despite vaccination?
Response:
We appreciate this observation Most participants received the standard two-dose schedule. Participants were recruited as already vaccinated individuals, and to exclude prior SARS-CoV-2 infection despite vaccination, serological and cellular testing and medical history review were performed. We have clarified this in the Supplementary file.
Comment 3:
«A reminder of the composition of the vaccines would be welcomed. »
Response:
We agree with the Reviewer and have added a short description of the composition of each vaccine (vector-based, inactivated, or peptide-based) in the Materials and Methods section (p. 5, lines 205–209).
Comment 4:
In the discussion two points have to be included:
- How does the ID test compare to anti-N antibody tests presently in use for diagnosis of past infection?
- Would the ID test remain positive over a long period? The cellular (CD8+) immunity may last for a long time (e.g., see Front Microbiol. 2022;12:803031).
Response:
We thank the Reviewer for these important suggestions. In the revised Discussion (p. 15, lines 515–527), we have added:
- A comparison of the diagnostic value of the ID test with anti-N antibody assays need to be clarified in future studies.
- A discussion on the persistence of T-cell responses, citing the suggested reference and related literature, and emphasizing the need for longitudinal studies to assess the durability of IDR test positivity.
Comment 5:
Could the “false positives” be related to other human coronavirus infections?
Response:
We appreciate this observation. Indeed, cross-reactivity with endemic human coronaviruses cannot be fully excluded. We have added a note on this possibility in the Discussion (p. 15, lines 515–527) and suggested future studies to evaluate cross-reactivity more directly.
Comment 6:
Among infections with variants, the IDR test appears less efficient. Any comments?
Response:
We thank the Reviewer for raising this point. Based on analyses using DeLong's test, we did not observe statistically significant differences in the intradermal test results among volunteers who had recovered from COVID-19 caused by different viral variants (Table 3).
Comment 7:
Line 19: Give the name of the virus in full and indicate its official name (Betacoronavirus pandemicum).
Response:
We have corrected the manuscript to provide the full official name of the virus at first mention (p. 2, line 48).
Comment 8:
The bacteria names (Salmonella typhimurium, Escherichia coli) have to be written in italics.
Response:
We thank the Reviewer for pointing this out. All bacterial species names have been revised and are now consistently italicized in the manuscript.
Comment 9:
Table 1: “coagulogram, total IgE” — the line is empty?
Response:
We acknowledge this oversight. We have corrected Table 1 to clarify the information related to coagulogram and total IgE. The table has been updated in the revised version.
Comment 10:
Figure 1: yellow is for M protein?
Response:
We appreciate this remark. Figure 1 has been revised, and both the overall figure and the embedded text/labels have been enlarged to improve legibility. The updated figure is included in the revised manuscript.
We thank the Reviewer once again for the detailed and constructive comments, which have helped us to clarify methodological details, strengthen the discussion, and improve the accuracy of the manuscript.
Round 2
Reviewer 3 Report
Comments and Suggestions for Authors
Thank you, it was a pleasure to review this manuscript which has thoroughly addressed my previous concerns, and I am now pleased to recommend publication.
Author Response
We sincerely thank the Reviewer for the positive feedback and kind recommendation. We appreciate the constructive comments and suggestions provided during the review process, which have improved the quality and clarity of our manuscript.
Reviewer 4 Report
Comments and Suggestions for Authors
The comments have been taken into account. The official name of the SARS-Cov-2 has to be added on line 62-63: Betacoronavirus pandemicum.
After that, the paper is acceptable.
Author Response
We thank the Reviewer for this valuable comment. The official name of SARS-CoV-2 has been added at line 49 in the revised manuscript.